

# Size matters but hunger prevails—begging and provisioning rules in blue tit families

Nolwenn Fresneau, Arne Iserbyt, Carsten Lucass and Wendt Müller

Department of Biology/Behavioural Ecology and Ecophysiology Group, Universiteit Antwerpen, Wilrijk, Antwerp, Belgium

## ABSTRACT

It is commonly observed in many bird species that dependent offspring vigorously solicit for food transfers provided by their parents. However, the likelihood of receiving food does not only depend on the parental response, but also on the degree of sibling competition, at least in species where parents raise several offspring simultaneously. To date, little is known about whether and how individual offspring adjusts its begging strategy according to the entwined effects of need, state and competitive ability of itself and its siblings. We here manipulated the hunger levels of either the two heaviest or the two lightest blue tit (*Cyanistes caeruleus*) nestlings in a short-term food deprivation experiment. Our results showed that the lightest nestlings consistently begged more than the heaviest nestlings, an effect that was overruled by the tremendous increase in begging behaviour after food deprivation. Meanwhile, the amplified begging signals after food deprivation were the only cue for providing parents in their decision process. Furthermore, we observed flexible but state-independent begging behaviour in response to changes in sibling need. As opposed to our expectations, nestlings consistently increased their begging behaviour when confronted with food deprived siblings. Overall, our study highlights that individual begging primarily aims at increasing direct benefits, but nevertheless reflects the complexity of a young birds' family life, in addition to aspects of intrinsic need and state.

## INTRODUCTION

Offspring begging behaviour is commonly thought to stimulate parents to provide care (*Godfray, 1991*; *Kilner & Johnstone, 1997*). Parents receive potentially cryptic information of offspring need or state, and are supposed to adjust their feeding strategy accordingly (*Kilner & Johnstone, 1997*; *Godfray & Johnstone, 2000*). This is empirically supported by experimental studies showing an increase in parental provisioning in response to an increase in offspring begging (*Mondloch, 1995*; *Dor & Lotem, 2010*; *Royle, Smiseth & Kölliker, 2012*). However, offspring begging signals of need may no longer induce a parental response in unpredictable and poor environments, as parents in such conditions may preferentially feed offspring that signal good condition or quality (*Caro et al., 2016*). Parent-offspring communication may further be complicated by an evolutionary

Corresponding author
Arne Iserbyt,
arne.iserbyt@uantwerpen.be

conflict of interest between parents and their young regarding the amount of provided food, with nestlings asking for more than is optimal for the parents to provide (*parent–offspring conflict, Trivers, 1974*). However, excessive offspring begging is constrained by the associated costs of displaying it, which may guarantee the honesty of the signal ('honest signalling theory': *Maynard Smith, 1991*; *Johnstone & Grafen, 1993*). Such costs include reduced offspring development (*Kilner, 2001*), increased predation risk (*Leech & Leonard, 1997*; *Haskell, 2002*) or physiological deterioration (*Moreno-Rueda, 2010*; *Noguera et al., 2010*; *Moreno-Rueda & Redondo, 2012*; *Soler et al., 2014*). Given these fitness costs of begging, offspring are expected to optimize the intensity, duration and timing of their displays in order to maximize their gains.

However, receiving care does not only depend on the parents, at least not if an offspring is competing with sibling(s). Each offspring is expected to behave selfishly within its brood as it shares at best only half of its genes with each of its siblings (*Trivers, 1974*; *Godfray, 1995a*; *Rodríguez-Gironés, Cotton & Kacelnik, 1996*; *Mock & Parker, 1997*). Thus, the likelihood of receiving care also depends on an individual's competitive ability, which is not necessarily equal among individuals of the same brood. In birds, for instance, hatching asynchrony creates a size difference among nestlings (*Cotton, Wright & Kacelnik, 1999*; *Rodríguez-Gironés, Zúñiga & Redondo, 2002*). Smaller nestlings may thus be outcompeted by their bigger and stronger nestlings, resulting in accumulated long-term need (*Oddie, 2000*; *Leonard, Horn & Parks, 2003*). Thus, the social environment that is shaped by all nestlings determines the outcome of competition and, therewith, the cost-benefit ratio of a given begging strategy. For example, the likelihood of receiving food may diminish when the nest is shared with hungry siblings. In this case, reducing begging effort may avoid the costs of unrewarded begging. In summary, each nestling is expected to adjust its begging behaviour according to both its own short- and long-term need, and according to the intertwined effects of state, need and competitive abilities of its siblings (*Price, Harvey & Ydenberg, 1996*; *Price, Ydenberg & Daust, 2002*). Experimental studies testing these predictions are nearly absent from the scientific literature.

We here investigated whether and how blue tit (*Cyanistes caeruleus*) nestlings adjust their begging behaviour according to their own and their siblings' hunger level, taking the hierarchical position within the brood into account. Specifically, we manipulated the hunger level of two target nestlings during a short-term (90 min) food deprivation experiment. These two nestlings were the extreme positions within the brood hierarchy (i.e. either the two heaviest or the two lightest nestlings), which are supposed to differ in competitive ability and long-term need (*Price, Harvey & Ydenberg, 1996*; *Lotem, 1998*; *Cotton, Wright & Kacelnik, 1999*). We predict that increased begging intensity by food-deprived nestlings should be accompanied by a decreased begging intensity from their siblings. Furthermore, we expect that the lightest siblings will be more reluctant to withdraw from competition as their long-term need and, therefore, their intrinsic motivation to access food, would be higher than the intrinsic motivation of heavier nestlings. Under the assumption that begging, among others, is an honest signal of need

in blue tits, we expect a positive parental response to nestlings with experimentally exaggerated begging behaviour.

## METHODS

This experiment took place in May 2014 in a nest-box population of blue tits breeding in Peerdsbos, a mature oak-beech forest in the north of Antwerp (N51°16′, E4°28′, Belgium). Nest boxes were checked daily to determine laying date, clutch size, the onset of incubation and hatching date (here defined as day 1). All nestlings within a nest hatched within 24 h. Nestling body mass was determined on day 12. To enable individual recognition on the video recordings, the two heaviest and the two lightest nestlings were marked with respectively a horizontal and vertical line on the upper mandible of the beak, using a non-toxic black marker (Artline®70N). We are confident that this marking has very limited effect on parental behaviour, because the markings become invisible from the parents' perspective when nestlings open their beak when soliciting for food and thus presenting gape and mouth flange colouration (*Heeb, Schwander & Faoro, 2003*). The difference in body mass between the two heaviest and the two lightest nestlings was on average 1.74 ± 0.22 g (mean ± SD). Infrared cameras (420TVL) were installed inside the nest-box underneath the lid, facing downwards. This was done one day before the experiment, so that the adults were already habituated to the camera (*Lucass et al., 2016*). The entire experiment was video monitored and started by closing the entrance of the nest box with small iron bars for 30 min (Fig. 1), which prevented parents from entering. This was done to reduce the behavioural bias due to feeding events prior to the behavioural measurements, and thus to make hunger levels comparable within and across nests (*Iserbyt et al., 2017*). The nest entrance was then reopened for another 30 min to monitor the parental feeding and nestling begging duration. This period is referred to as the 'Control' period before the food deprivation experiment (Fig. 1).

We then removed the two heaviest nestlings or the two lightest nestlings and food deprived (FD) them for 90 min. We kept both nestlings in a cloth bag in an insulated box along with a hand warmer. We replaced these nestlings with foster nestlings of similar mass from a neighbouring nest box, to keep the brood size constant during the food deprivation period of the two focal nestlings. Which nestlings (the two heaviest or lightest nestlings, respectively) were selected for the food deprivation procedure was alternated across nests ($N_{\text{lightest FD}}$ = 15 nests; $N_{\text{heaviest FD}}$ = 15 nests). Brood size of the experimental nests varied from 8 to 14 nestlings and did not differ between both experimental groups.

After 90 min of food deprivation the focal nestlings were returned to their nest, and the foster nestlings were returned to their own nest. We then repeated the procedure as detailed in the control period (now 'After FD' in Fig. 1), i.e. parental feeding and nestling begging duration was monitored after a 30-min period with the nest entrance closed. This period was again necessary to standardize hunger levels of the non-FD nestlings. The experiment was carried out in agreement with Belgian and Flemish legislation and was approved by the Ethical Committee for animals (ECD) of the University of Antwerp (license number 2011-10).

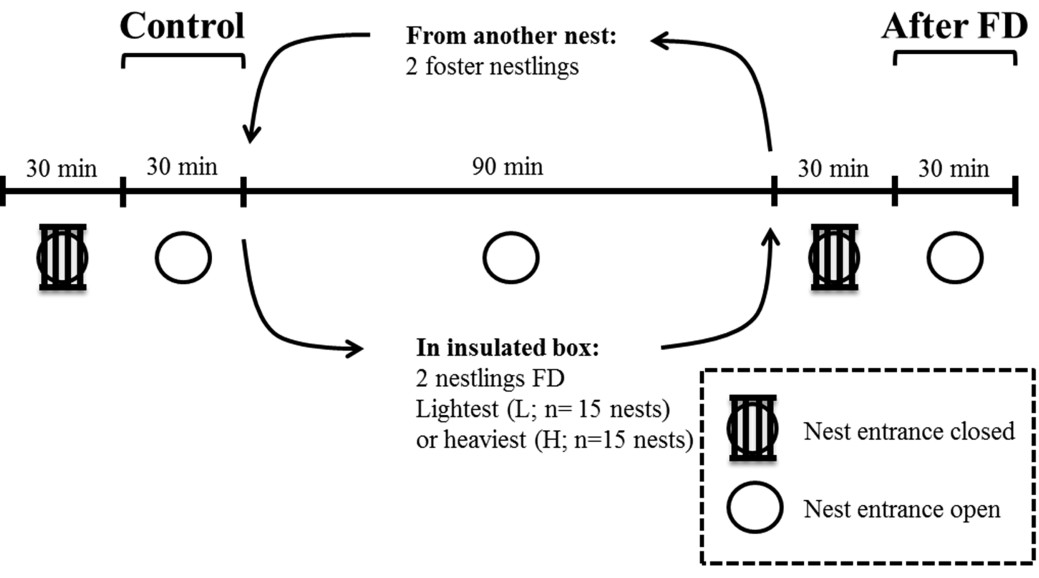

**Figure 1 Overview and timeline of the experimental design when nestlings were 12 days old.** FD indicates that either the two heaviest or two lightest nestlings are food deprived for 90 min.

## Video analyses

All videos were analysed blinded and by the same person, using video analysis software (NOLDUS Observer XT 10.0, Wageningen, The Netherlands). We quantified begging behaviour by measuring the nestling gaping time, i.e. the time when the beak of the nestling is open. This begging duration was measured from the moment that a parent entered the nest box, until the moment that one of the nestlings was fed. In blue tits, parents typically provide one food item at a time, and begging rapidly ceases once that item has been transferred. Furthermore, only the first two feeding visits were considered, to assure that the brood was still at the experimentally manipulated hunger level. Note that in case of the food deprivation each time only two nestlings were FD, either the two heaviest or the two lightest nestlings. Unfortunately, it was not possible to distinguish individual nestlings within its body mass rank during the video analyses. For each feeding visit, behavioural data from both nestlings within the mass rank were therefore summed and analytically considered as one individual (RankID, see below). This results in eight ($2^3$) behavioural observations per nest; i.e. two feeding events, two mass ranks (light and heavy) and two periods (control and after FD; see Supplementary Material). During provisioning, we scored which nestling received the food item (one of both light, one of both heavy nestlings, or neither of the two groups; 1 when fed and 0 when not fed).

## DATA ANALYSES

The begging duration was log10 transformed to meet assumptions of normality and centred by conversion to standardized ($z$) scores. However, we present mean values of the raw data (in seconds ± standard error) whenever useful. In four videos
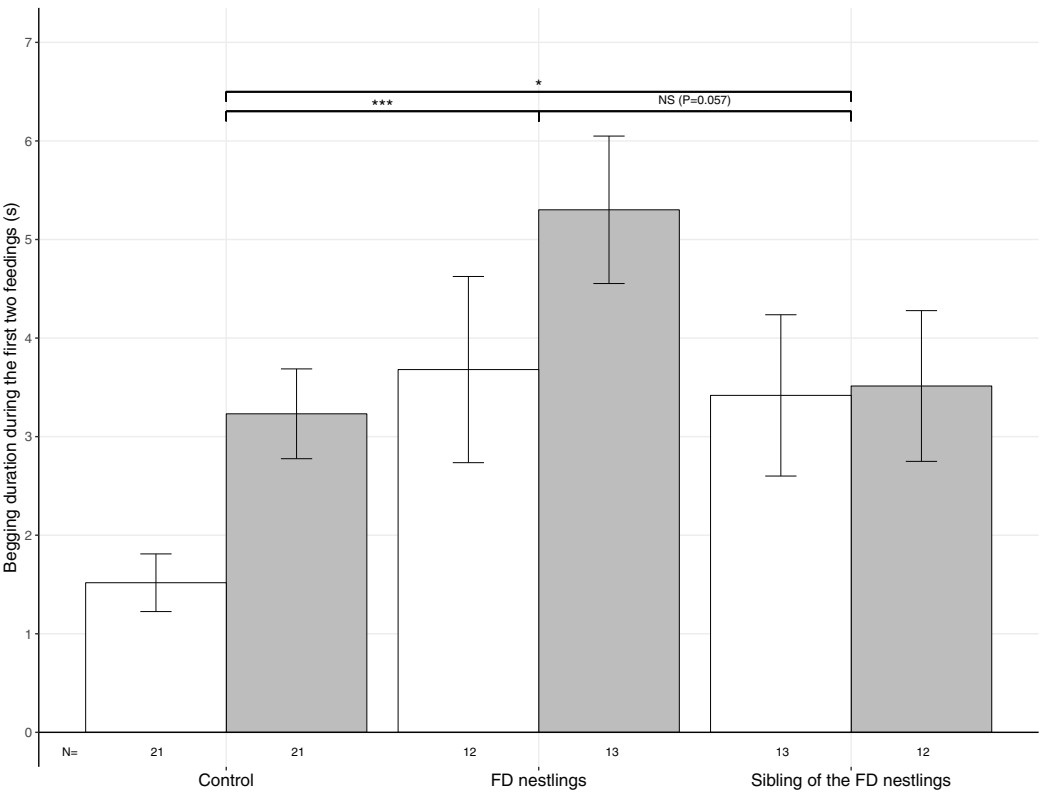

**Figure 2 Mean (±SE) begging duration according to context and body mass rank.** The three contexts are Control (no nestlings are food deprived), FD nestlings (after 90 min of food deprivation) and siblings of the 90 min FD nestlings. Average values of the two heaviest and the two lightest nestlings are represented as respectively white and grey histograms. Post hoc comparisons across contexts (heaviest and lightest nestlings combined) are represented with asterisks (*$P < 0.05$; ***$P < 0.001$). $N$ represents the number of experimental nests.

(three Controls and one FD), neither of the parents returned to the nest within the 30 min of video recording and were therefore excluded from the analysis.

A linear mixed effect (LME) model was used to test whether nestling begging duration during the first two parental feeding bouts varied according to nestling context, nestling body mass rank (heaviest or lightest), their interaction and brood size. The three possible nestling contexts were 'Control' (represents the individual status prior to the 90 min FD), 'FD nestling' (status after 90 min FD) and 'Sibling of the FD nestling' (also after 90 min FD; Figs. 2 and 3). RankID (unique ID of the heaviest or lightest two nestlings within a nest) nested in NestID was included as a random effect to account for pseudoreplication of non-independent data.

The probability of receiving food was analysed in a generalized linear mixed effect (GLME) model with the variable 'fed' (1 = fed and 0 = not fed) as a response variable using a binomial distribution and a clog-log link (since there were more 0 than 1). RankID nested within NestID was used as a random effect and nestling context, rank, their interaction and brood size were included as explanatory variables.

In all cases, we performed stepwise backward model selection procedures starting from the full model. We tested fixed effects in the models fitted with the maximum

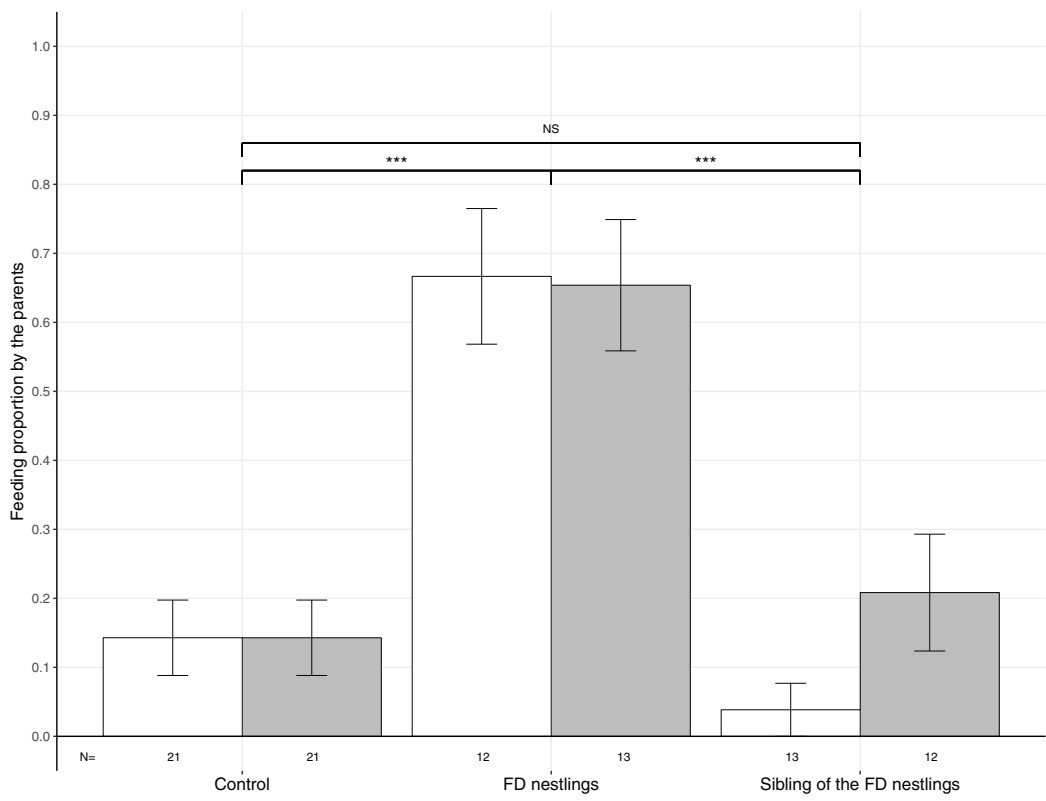

**Figure 3 Mean (±SE) proportion of parental feeds according to context and body mass rank.** Parental food transfers could be directed towards the two heaviest (white histograms), the two lightest (grey histograms) or non-focal nestlings. The contexts are Control (no nestlings are food deprived), FD nestlings (after 90 min of food deprivation) and siblings of the 90 min FD nestlings. Post hoc comparisons across contexts (heaviest and lightest nestlings combined) are represented with asterisks (***P < 0.001). N represents the number of experimental nests.               

likelihood by comparing a model with and without the fixed effect in question using likelihood ratio tests (LRT) against a $\chi^2$ distribution. Non-significant fixed effects (P value greater than 0.05) were removed one by one from the model starting with the least significant. For all variables not selected in the final model we provide the $\chi^2$ value and the P value associated with the model comparison analysis in the results section. The final model was fitted with restricted maximum likelihood to obtain the coefficients for the fixed effects and variance estimates for the random effects (*Zuur et al., 2009*). All statistics were performed in R version 2.15.2 (*R Development Core Team, 2012*), using the nlme package (*Pinheiro et al., 2013*) and the lme4 package (*Bates et al., 2015*).

# RESULTS

## Nestling begging

Begging duration significantly differed according to the nestling context (Table 1). Nestlings almost doubled their begging duration following food deprivation (mean ± SE; Control: $N = 84$, 2.37 ± 0.28 s; FD nestlings: $N = 50$, 4.52 ± 0.60 s; post hoc test: $\chi^2 = 16.79$, $P < 0.001$; Fig. 2). Interestingly, siblings of the FD nestlings significantly increased their begging duration compared with the non-food deprived control period (siblings
**Table 1 Model comparisons exploring variation in nestling begging and parental provisioning behaviour.**

| Effect | df | AIC | LRT | P |
|---|---|---|---|---|
| **(a) Nestling begging duration** | | | | |
| Rank X context | 2.128 | 512.5 | 3.5 | 0.17 |
| **Brood size** | **1.24** | **512.0** | **7.45** | **0.006** |
| **Rank** | **1.25** | **517.4** | **11.95** | **<0.001** |
| **Context** | **2.128** | **527.4** | **16.56** | **<0.001** |
| None | | 539.3 | | |
| **(b) Parental provisioning** | | | | |
| Brood size | – | 181.0 | 0.01 | 0.93 |
| Rank X context | – | 180.0 | 3.39 | 0.18 |
| Rank | – | 178.8 | 0.38 | 0.54 |
| **Context** | **–** | **220.7** | **45.8** | **<0.001** |
| None | – | 178.8 | – | – |

Notes:
These models test the influence of the nestling context (Control, FD and Sibling FD), nestling body mass rank (heavy and light) and brood size on nestling begging duration (LME, a) and parental provisioning behaviour (GLME, b) during the first two parental feeding bouts. Rank ID nested in Nest ID was included as a random effect. Each AIC value is based on a model that includes the respective variable, in addition to the variables situated under it in the table. Significant variables that were retained in the reduced model are highlighted in bold.

of the FD nestlings: $N = 50$, 3.46 ± 3.93 s; $\chi^2 = 4.03$, $P = 0.045$; Fig. 2). FD nestlings tended to beg longer than their siblings did at the same moment ($\chi^2 = 3.61$, $P = 0.057$; Fig. 2). Nestling begging duration was also influenced by the body mass rank, with the lightest nestlings begging significantly longer than the heaviest nestlings (lightest: $N = 42$, 3.89 ± 0.36; heaviest: $N = 42$, 2.62 ± 0.37), an effect that was consistent across contexts (Table 1; Fig. 2). Finally, begging duration significantly decreased with brood size (estimate ± SE: −0.18 ± 0.06; Table 1).

## Parental provisioning

Parents significantly adjusted their feeding strategy according to the nestling context (Table 1; Fig. 3). FD nestlings received remarkably more prey items than their siblings (respectively $N = 50$, 66.0%; and $N = 50$, 12.0%; post hoc test: $z = 4.52$, $P < 0.001$). Siblings did not receive more food than before the food deprivation experiment ($N = 84$, 14.3%; $z = 0.36$, $P = 0.72$). Neither the body mass rank of the nestlings nor the brood size influenced parental feeding decisions (Table 1; Fig. 3).

## DISCUSSION

### Begging in function of need and state

Ninety minutes of food deprivation drastically increased begging duration of blue tit nestlings. Thus begging, as measured in this study, does carry information about short-term nutritional need (*Godfray, 1991*), which also supports the concept of begging being a plastic trait (*Kedar et al., 2000*). Furthermore, we found that the lightest two nestlings consistently begged longer than the two heaviest nestlings (*Price, Harvey & Ydenberg, 1996*;

*Bonisoli-Alquati et al., 2011*), even when hunger levels are comparable among nestlings. Such state-dependent begging behaviour has been interpreted as a signal of long-term need (*Price, Harvey & Ydenberg, 1996*). The lightest nestlings may have suffered from an enduring backlog demand, because they were persistently outcompeted by their heaviest siblings (*Smith & Montgomerie, 1991*; *Dearborn, 1998*) or, alternatively, because parents made the choice to feed better quality nestlings (but see further; *Kilner, 1995*; *Moreno-Rueda et al., 2007*).

## Begging in function of the social environment

Our main aim was to test whether and how nestlings adjust their signalling to aspects of their social environment, that is their position in the body mass hierarchy and the hunger level of their siblings. We found that non-food deprived nestlings only adjusted their begging level according to the elevated hunger levels of their siblings, and not according to their own or their sibling's rank in the hierarchy. This result is in contrast to a previous studies reporting that less needy chicks withdraw from competition if the chance of obtaining food and thus the cost-benefit ratio worsens (*Romano et al., 2012*). In this scenario, siblings of FD nestlings may gain indirect benefits from their altruistic behaviour. Nestlings are therefore expected to negotiate with their siblings about who should receive the next prey item (*Roulin, Kölliker & Richner, 2000*). However, this hypothesis is not supported by our findings, as we found the opposite pattern in our study system. Specifically, siblings increased their begging when the two focal nestlings were FD, although this did not affect the likelihood to obtain food. It is possible that we did not find evidence for negotiation or altruistic behaviour because brood sizes of blue tits are larger (8–14 nestlings, in our population) than in barn owls (*Tyto alba*; two to nine nestlings; *Roulin, Kölliker & Richner, 2000*) and barn swallows (*Hirundo rustica*; three to seven nestlings; *Romano et al., 2012*), in which sibling negotiation has been reported. Scramble competition may increase with the number of siblings and this makes it harder for each individual to receive food (*Smith & Montgomerie, 1991*; *Leonard et al., 2000*). Thus, withdrawing from competition during a feeding event in large broods may not substantially increase the chance of being fed in the next feeding event.

## Parental response

Food deprived nestlings begged vigorously, which largely improved the chance to be fed compared with their nest mates. This indicates that parents respond positively to begging and thus short-term need, which is (among others) a prerequisite for begging to evolve as an evolutionary stable strategy in the case of honest signalling models (*Godfray, 1995b*). This finding is in line with a number of previous studies (*Cotton, Kacelnik & Wright, 1996*; *Price, Harvey & Ydenberg, 1996*; *Lotem, 1998*), and supports the function of offspring begging both in a signalling context as well as in sibling competition (*Royle, Hartley & Parker, 2002*). However, the lightest nestlings, which on average begged more, did not receive more food. Our observed parental response is in line with previous findings, indicating that younger nestlings beg more than their older siblings but receive less or 'only' equal amounts of food (*Lotem, 1998*; *Cotton, Wright & Kacelnik, 1999*).

This can be interpreted as a brood reduction strategy, given that hatching asynchrony permits reduction of the cost of rearing marginal offspring and ultimately, in case of food shortage, to reduce the brood size (*Forbes & Glassey, 2000*).

## CONCLUSIONS

Blue tit nestlings adjusted their begging in function of their own short-term need. They additionally altered their begging according to (changes in) their social environment, i.e. the enhanced hunger level of their nest mates. However, blue tit nestlings never withdrew from competition in favour of needier siblings, despite their genetic relatedness and potential indirect fitness benefits they could have gained from it. On the contrary, overall brood begging levels rather increased when a subset of nestlings was food-deprived, suggesting higher within-brood sibling competition which matches the assumptions of scramble competition models. Furthermore, we showed that the position within the body mass hierarchy had a large influence on individual begging behaviour, but that the behavioural adjustments to changes in need and the social environment itself are state-independent. Ultimately, blue tit parents fed the hungriest nestling, which was the one begging the most, independent of whether it was the heaviest or lightest nestling within the brood. Taken together, individual begging strategies are fine-tuned to the complexity of a young birds' family life, but primarily vary with intrinsic need.

## ACKNOWLEDGEMENTS

The authors wish to thank William and Petra Van Dieren and the Agentschap Natuur en Bos (ANB) for providing electricity and facilities during field work. We thank Erik Matthysen for giving access to his field site, Joris Elst for help with the practical work, April Ward for analysing videos and Jamie MacLaren for improving the writing. The authors also wish to thank Gregorio Moreno-Rueda, Douglas W. Mock and two anonymous reviewers for their comments on a previous version of this manuscript.

### Funding

This study was financially supported by the University of Antwerp (DOCPRO4, ID: 27332 to Wendt Müller) and FWO Flanders (project ID: 1517815N and 12I1916N to Arne Iserbyt). The funders had no role in study design, data collection and analysis, decision to publish, or preparation of the manuscript.

### Grant Disclosures

The following grant information was disclosed by the authors:
University of Antwerp: DOCPRO4, ID: 27332.
FWO Flanders: 1517815N and 12I1916N.

### Competing Interests

The authors declare that they have no competing interests.

## Author Contributions

- Nolwenn Fresneau conceived and designed the experiments, performed the experiments, analyzed the data, prepared figures and/or tables, authored or reviewed drafts of the paper, approved the final draft.
- Arne Iserbyt prepared figures and/or tables, authored or reviewed drafts of the paper, approved the final draft.
- Carsten Lucass performed the experiments, authored or reviewed drafts of the paper, approved the final draft.
- Wendt Müller conceived and designed the experiments, contributed reagents/materials/analysis tools, authored or reviewed drafts of the paper, approved the final draft.

## Animal Ethics

The following information was supplied relating to ethical approvals (i.e. approving body and any reference numbers):

The experiment was carried out in agreement with Belgian and Flemish legislation and was approved by the Ethical Committee for animals (ECD) of the University of Antwerp (license number 2011-10).

## Data Availability

The raw measurements that are used for the analyses are provided in File S1.

## Supplemental Information

Supplemental information for this article can be found online at http://dx.doi.org/10.7717/peerj.5301#supplemental-information.

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
