# Peer review of "Size matters but hunger prevails—begging and provisioning rules in blue tit families"

_PeerJ, doi:10.7717/peerj.5301_

## Round 0.1 · original submission · Major Revisions

Dear authors
as you can see from comments of reviwer 1 your ms needs a thorough revision. Try to consider these arguments in your revision as we will send therevision to the same reviewers.
Greetings
Michael Wink
Academic editor

Reviewer 1 ·

Basic reporting

This manuscript investigates several factors that may influence begging in blue tits, namely hunger, size rank, and their siblings' begging. The manuscript is well-written. The figures and table are clear and helpful. The raw data and R file are clearly presented. It has self-contained results that are relevant results to hypotheses.

I believe it could have more background/context for the field. For instance, [33-35] Not all empirical / experimental studies show an increase in parental provisioning in response to increased offspring begging (see, for example, Caro et al 2016's meta-analysis of begging and provisioning, which found that parental responsiveness to begging varies by species and wherein you can find examples of studies showing no change in parental provisioning due to begging). Your sentence, as written, implies that this is a settled and consistent finding. I think the introduction could be strengthened by including some of the nuance around this signalling system.

[76-77] Again, the function of begging is not as settled in the literature as it might seem based on your introduction. You write under the assumption that begging is an honest signal of need to parents, yet it is also possible that it is a signal of quality, a signal of hunger, scramble competition (a non-signal), or a signal to nestmates rather than to parents (for example, Grafen 1990; Royle et al 2002; Mock et al 2011; Rodriguez-Girones et al 1996 and 1998; Roulin et al 2000 etc). It is alright to write under the honest signal of need framework, but you should acknowledge that other possibilities exist in the introduction, and not just the discussion.

The discussion has sufficient literature references, and was very well written. I think it situated the current study well within the field and ongoing research. It also summarized your findings succinctly and clearly.

Experimental design

The research question is well defined, relevant, and meaningful. I have no ethical concerns about the experiment.

Overall, the experimental design of the study was sound, and I commend the authors on the food deprivation experiment and video analysis.

The methods may need to be clarified (see next section).

Validity of the findings

Issues with data / analysis, or with how methods are written:

*Random effects:
If I am interpreting your excel sheet correctly, nestling ID is a combination of nest + weight class. So I believe you have coded 4 different chicks as only 2 different chicks (for example, both light chicks in nest 16 are coded as 16L, and both heavy chicks are coded as 16H). Therefore, you are not really controlling for chick ID when you include ID, but instead you are controlling for weight within brood, which is also included in your fixed effects. This means the true effect of weight class on begging, and the true effect of individual ID, cannot be determined by your current analysis. I assume that it was impossible to distinguish between the two chicks in each weight class based on the beak marking [84-86]. You could help fix this by including only nest as a random effect in the begging analysis, since controlling for chick ID seems impossible. You should then also write in the methods section that you include two observations per chick in analyses, but were unable to control for chick ID.

You should also control for feeding event (FeedingNr?) for both the begging and parental response analyses, since data from multiple chicks on the same feeding event are not independent of each other, and it is pseudoreplication not to account for this non-independence.

*Experimental treatment:
I am also confused by the "context" variable in your excel sheet. Based on your methods [96-103], you food deprived both or neither chick in a weight class. You also coded the first two feeding visits per nest [118-119]. This means that in every feeding visit, there should be two food deprived nestlings. However, in your data sheet, there is only one food-deprived nestling per feeding visit. Either your methods section needs to be rewritten so it is clear what you did experimentally and analytically, or there is a problem with the context variable in your data sheet (which undermines any analysis of the effect of food deprivation).
* * *
Given the analyses presented, the conclusions are well stated, linked to the original research question, and interpreted appropriately.

Additional comments

[Throughout] I think you clarified the potential distinctions between short and long term need well.

[22-23] How can you be sure that begging signals were the "only" cue parents used? They might have used something that you didn't measure and analyse.

[48] "Selfish" should be "selfishly"

[49] Missing the word "siblings" after "its"

[66] "Siblings" requires an apostrophe

[75] Missing a comma after "food"

[95] Is the control period only from before food deprivation?

[107] "minutes" should be "minute"

[112] If your video analysis was blinded, please include that here.

[140] Either put a space before and after the "=", or don't have spaces at all.

[146] "Greater than" would be better than "superior to"

[Figure 2] Space missing in "andbody"

[Figure 3] Space missing in "massrank"

Reviewer 2 ·

Basic reporting

see general comments.

Experimental design

see general comments.

Validity of the findings

see general comments.

Additional comments

Fresneau et al ms Size matters but hunger prevails – PEERJ
This is a cleverly designed, thoroughly researched experimental study. Basic reporting, experimental design, and validity of findings all meet the standards and scope of PEERJ. Well written text, conclusions well stated.
One question concerning methods: Any indication, that marking nestlings´ beak with black colour influenced parental behaviour?

---

## Round 0.2 · accepted · Accept

Dear authors

Your revision is adequate. Congratulations, we can now accept your ms. Thanks for publishing with us.

Kind regards,

Michael Wink
AE

# Reviewer 1 ·

Basic reporting

See general comments

Experimental design

See general comments

Validity of the findings

See general comments

Additional comments

I am satisfied with the changes made for this revised manuscript, and I support its publication. The data structure and methods are much clearer now, and the statistical analyses seem sound.